# GeoSeq2Seq: Information Geometric Sequence-to-Sequence Networks

**Alessandro Bay**
Cortexica Vision Systems Ltd.
London, UK
`alessandro.bay@cortexica.com`

**Biswa Sengupta**
AI Theory Lab (Noah's Ark)
Huawei Technologies R&D, London, UK
`biswa.sengupta@huawei.com`

## Abstract

The Fisher information metric is an important foundation of information geometry, wherein it allows us to approximate the local geometry of a probability distribution. Recurrent neural networks such as the Sequence-to-Sequence (Seq2Seq) networks that have lately been used to yield state-of-the-art performance on speech translation or image captioning have so far ignored the geometry of the latent embedding, that they iteratively learn. We propose the information geometric Seq2Seq (GeoSeq2Seq) network which abridges the gap between deep recurrent neural networks and information geometry. Specifically, the latent embedding offered by a recurrent network is encoded as a Fisher kernel of a parametric Gaussian Mixture Model, a formalism common in computer vision. We utilise such a network to predict the shortest routes between two nodes of a graph by learning the adjacency matrix using the GeoSeq2Seq formalism; our results show that for such a problem the probabilistic representation of the latent embedding supersedes the non-probabilistic embedding by 10-15%.

## 1 Introduction

Information geometry situates itself in the intersection of probability theory and differential geometry, wherein it has found utility in understanding the geometry of a wide variety of probability models (Amari & Nagaoka, 2000). By virtue of Cencov's characterisation theorem, the metric on such probability manifolds is described by a unique kernel known as the Fisher information metric. In statistics, the Fisher information is simply the variance of the score function. In Bayesian statistics, it has found utility in terms of Riemannian Markov Chain Monte Carlo (MCMC) methods (Girolami & Calderhead, 2011) while for computer vision it has resulted in the Fisher kernel encoding (Perronnin & Dance, 2006). Practitioners have also used the geometric make-up of feature vectors obtained from a deep convolutional neural network (dCNN) to rank images (Qian et al., 2017) by encoding them using Fisher kernels. Apart from traditional signal processing methodologies like Kalman filters or Hidden Markov Models, recurrent neural networks that have proved to be beneficial for sequential data haven't quite utilised the geometry of the latent structure they learn.

There are two paths of an intersection of recurrent networks with Riemann geometry. The first lies in using the natural gradient to optimize loss functions of deep neural networks (Pascanu & Bengio, 2013). This affords invariance to the optimization procedure by breaking the symmetry in parameter space. The other is utilizing the geometry of the latent space to augment classification accuracy. In this paper, we combine a specific sort of recurrent network – the Sequence-to-Sequence (Seq2Seq) model – and utilize the Riemann geometry of the embedded space for boosting the performance of the decoder. We test the algorithm on a combinatorially hard problem called the *shortest route problem*. The problem involves a large graph wherein the shortest route between two nodes in the graph are required. Specifically, we use a meta-heuristic algorithm (a vanilla $A^*$ algorithm) to generate the shortest route between two randomly selected routes. This then serves as the training set for our GeoSeq2Seq network.

## 2 RELATED WORKS

Recently, the research direction of combining deep learning with methods from information geometry has proven to be an exciting and a fertile research area. In particular, natural gradient methods in deep learning have recently been explored to model the second-order curvature information. For example, Natural Neural Networks (Desjardins et al., 2015) have sped up convergence by adapting their internal representation during training to improve the conditioning of the Fisher information matrix. On the other hand, it is also possible to approximate the Fisher information matrix, either with a Gaussian graphical model, whose precision matrix can be computed efficiently (Grosse & Salakhudinov, 2015) or by decomposing it as the Kronecker product of small matrices, which capture important curvature information (Grosse & Martens, 2016).

More closely related to the topic of this paper, Fisher vector encodings and deep networks have been combined for image classification tasks (Sydorov et al., 2014). For example, Fisher vector image encoding can be stacked in multiple layers (Simonyan et al., 2013b), showing that convolutional networks and Fisher vector encodings are complementary. Furthermore, recent work has introduced deep learning on graphs. For example, Pointer Networks (Vinyals et al., 2015) use a Seq2Seq model to solve the travelling salesman problem, yet it assumes that the entire graph is provided as input to the model.

To the best of our knowledge, there has been very little work on using the Fisher vectors of a recurrent neural encoding, generated from RNNs (recurrent neural networks), LSTMs (Long short-term memory) and GRUs (Gated recurrent units) based sequence-to-sequence (Seq2Seq) models. Therefore, the work presented here is complementary to the other lines of work, with the hope to increase the fidelity of these networks to retain the memory of long sequences.

## 3 METHODS

In this section, we describe the data-sets, the procedure for generating the routes for training/test datasets, and the deployment of information geometric Sequence-to-Sequence networks that forms the novel contribution of this paper. All of the calculations were performed on a i7-6800K CPU @ 3.40GHz workstation with 32 GB RAM and a single nVidia GeForce GTX 1080Ti graphics card.

### 3.1 DATASETS

The graph is based on the road network of Minnesota[1]. Each node represents the intersections of roads while the edges represent the road that connects the two points of intersection. Specifically, the graph we considered has 376 nodes and 455 edges, as we constrained the coordinates of the nodes to be in the range $[-97, -94]$ for the longitude and $[46, 49]$ for the latitude, instead of the full extent of the graph, i.e., a longitude of $[-97, -89]$ and a latitude of $[43, 49]$, with a total number of 2,642 nodes.

### 3.2 ALGORITHMS

#### THE $A^*$ META-HEURISTICS

The $A^*$ algorithm is a best-first search algorithm wherein it searches amongst all of the possible paths that yield the smallest cost. This cost function is made up of two parts – particularly, each iteration of the algorithm consists of first evaluating the distance travelled or time expended from the start node to the current node. The second part of the cost function is a heuristic that estimates the cost of the cheapest path from the current node to the goal. Without the heuristic part, this algorithm operationalises the Dijkstra's algorithm (Dijkstra, 1959). There are many variants of $A^*$; in our experiments, we use the vanilla $A^*$ with a heuristic based on the Euclidean distance. Other variants such as Anytime Repairing $A^*$ has been shown to produce superior performance (Likhachev et al., 2004).

Paths between two nodes selected uniformly at random are calculated using the $A^*$ algorithm. On an average, the paths are 19 hops long. The average fan-in/fan-out of a randomly selected node is 2.42.

---

[1]https://www.cs.purdue.edu/homes/dgleich/packages/matlab_bgl

We increase the combinatorial difficulty of the shortest route by not constraining the search to the local fan-out neighbourhood, rather the dimension of the search space is $n - 1$ with $n$ representing the number of nodes in the graph.

RECURRENT DEEP NETWORKS

We utilised Sequence-to-Sequence (Seq2Seq, Sutskever et al. (2014)) recurrent neural networks for the shortest route path prediction. Specifically, we use the following variants:

- An LSTM2RNN, where the encoder is modelled by a long short term memory (LSTM, Hochreiter & Schmidhuber (1997)), i.e.

$$i(t) = \text{logistic}\left(A_i x(t) + B_i h(t-1) + b_i\right)$$

$$j(t) = \tanh\left(A_j x(t) + B_j h(t-1) + b_j\right)$$

$$f(t) = \text{logistic}\left(A_f x(t) + B_f h(t-1) + b_f\right)$$

$$o(t) = \text{logistic}\left(A_o x(t) + B_o h(t-1) + b_o\right)$$

$$c(t) = f(t) \odot c(t-1) + i(t) \odot j(t)$$

$$h(t) = o(t) \odot \tanh\left(c(t)\right), \tag{1}$$

  while the decoder is a vanilla RNN (Goodfellow et al., 2016), i.e.

$$h(t) = \tanh(A x(t) + B h(t-1) + b), \tag{2}$$

  followed by a softmax output layer, i.e.

$$y(t) = \text{logsoftmax}(C h(t) + c), \tag{3}$$

  which gives the probability distribution on the following node, choosing it among the other $n - 1$ nodes.

- A GRU2RNN, where the encoder is modelled by a gated recurrent unit (GRU, Cho et al. (2014)), i.e.

$$z(t) = \text{logistic}\left(A_z x(t) + B_z h(t-1) + b_z\right)$$

$$r(t) = \text{logistic}\left(A_r x(t) + B_r h(t-1) + b_r\right)$$

$$\tilde{h}(t) = \tanh\left(A_h x(t) + B_h(r(t) \odot h(t-1)) + b_h\right)$$

$$h(t) = z(t) \odot h(t-1) + (1 - z(t)) \odot \tilde{h}(t), \tag{4}$$

  while the decoder is again a vanilla RNN with a softmax, as in Equations (2)-(3).

- An LSTM2LSTM, where both the encoder and the decoder are modelled by an LSTM as in Equations (1).

- A GRU2LSTM, where the encoder is a GRU (see Eqn. (4)) and the decoder is an LSTM (see Eqn. (1)).

- *GeoSeq2Seq*, our novel contribution, where the context vector obtained as in one of the previous models is further encoded using either Fisher vectors or vectors of locally aggregated descriptors (VLAD; see Figure 1), as described in the following section.

For all networks, the input is represented by the [source, destination] tuple (Figure 1), which is encoded in a context vector ($W$) and subsequently decoded into the final sequence to obtain the shortest path connecting the source to the destination. Moreover, during the test phase, we compute two paths, one from the source to the destination node and the other from the destination to the source node, that forms an intersection to result in the shortest path.

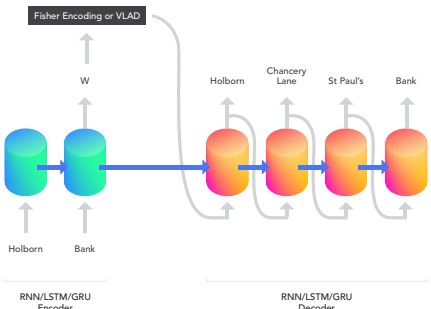

Figure 1: **Information geometric context vectors.** The context vectors (W) that are learnt using the recurrent neural networks are further encoded using either a Fisher vector based encoding or a Vector of Locally Aggregated Descriptors (VLAD) based encoding (see text). Such an encoded context vector is finally fed to a recurrent neural network based decoder.

INFORMATION GEOMETRY

Classical results from information geometry (Cencov's characterisation theorem; Amari & Nagaoka (2000)) tell us that, for manifolds based on probability measures, a unique Riemannian metric exists – the Fisher information metric. In statistics, Fisher-information is used to measure the expected value of the observed information. Whilst the Fisher-information becomes the metric ($g$) for curved probability spaces, the distance between two distributions is provided by the Kullback-Leibler (KL) divergence. It turns out that if the KL-divergence is viewed as a curve on a curved surface, the Fisher-information becomes its curvature. Formally, for a distribution $q(\psi)$ parametrised by $\theta$ we have,

$$
\begin{aligned}
\mathcal{KL}(\theta, \theta') &= \mathbb{E}_\theta\left[\log \frac{q(\psi\,|\theta)}{q(\psi\,|\theta')}\right] + \mathbb{E}_{\theta'}\left[\log \frac{q(\psi\,|\theta')}{q(\psi\,|\theta)}\right] \\
&= d\theta^T g(\theta) d\theta + \mathcal{O}(d\theta^3) \\
g_{ij}(\theta) &= \int_{-\infty}^{+\infty} q(\psi, \theta) \frac{\partial \ln q(\psi, \theta)}{\partial \theta_i} \frac{\partial \ln q(\psi, \theta)}{\partial \theta_j} d\psi \\
\mathcal{KL}[\theta_0 + \delta\theta : \theta_0] &\doteq \frac{1}{2} g_{ij}(\theta_0)(\delta\theta)^2.
\end{aligned}
$$

**Fisher encoding**   We use a Gaussian Mixture Model (GMM) for encoding a probabilistic sequence vocabulary ($W$) on the training dataset. The context vectors are then represented as Fisher Vectors (FV, Perronnin & Dance (2006)) – derivatives of log-likelihood of the model with respect to its parameters (the score function). Fisher encoding describes how the distribution of features of an individual context differs from the distribution fitted to the feature of all training sequences.

First, a set of $D$ dimension context vector is extracted from a sequence and denoted as $W = (w_1, ...w_i, ..., w_N : w \in \mathbb{R}^D)$. A $K$ component GMM with diagonal covariance is generated (Simonyan et al., 2013a; Cimpoi et al., 2015) on the training set with the parameters $\{\Theta = (\omega_k, \mu_k, \Sigma_k)\}_{k=1}^K$, only the derivatives with respect to the mean $\{\mu_k\}_{k=1}^K$ and variances $\{\Sigma_k\}_{k=1}^K$ are encoded and concatenated to represent a sequence $W(X, \Theta) = \left(\frac{\partial L}{\partial \mu_1}, ..., \frac{\partial L}{\partial \mu_K}, \frac{\partial L}{\partial \Sigma_1}, ..., \frac{\partial L}{\partial \Sigma_K}\right)$, where

$$
\begin{aligned}
L\left(\Theta\right) &= \sum_{i=1}^{N} log\left(\pi\left(w_i\right)\right) \\
\pi\left(w_i\right) &= \sum_{k=1}^{K} \omega_k N\left(w_i; \mu_k, \Sigma_k\right).
\end{aligned}
\tag{5}
$$

For each component $k$, mean and covariance deviation on each vector dimension $j = 1, 2...D$ are

$$
\begin{aligned}
\frac{\partial L}{\partial \mu_{jk}} &= \frac{1}{N\sqrt{\omega_k}} \sum_{i=1}^{N} q_{ik} \frac{w_{ji} - \mu_{jk}}{\sigma_{jk}} \\
\frac{\partial L}{\partial \Sigma_{jk}} &= \frac{1}{N\sqrt{2\omega_k}} \sum_{i=1}^{N} q_{ik} \left[\left(\frac{w_{ji} - \mu_{jk}}{\sigma_{jk}}\right)^2 - 1\right],
\end{aligned}
\tag{6}
$$

where $q_{ik}$ is the soft assignment weight of feature $w_i$ to the $k^{th}$ Gaussian and defined as

$$
q_{ik} = \frac{exp\left[-\frac{1}{2}\left(w_i - \mu_k\right)^T \Sigma_k^{-1}\left(w_i - \mu_k\right)\right]}{\sum_{t=1}^{K} exp\left[-\frac{1}{2}\left(w_i - \mu_t\right)^T \Sigma_t^{-1}\left(w_i - \mu_t\right)\right]}.
\tag{7}
$$

Just as the sequence representation, the dimension of Fisher vector is $2KD$, $K$ is the number of components in the GMM, and $D$ is the dimension of local context descriptor. After $l_2$ normalization on Fisher vector, the embedding can be learnt using an arbitrary recurrent neural network based decoder (Perronnin et al., 2010). In our experiments (see Section 4), as a proof of concept, we fixed the number of GMMs to $K = 1$, since more Gaussians would have increased the dimension acted upon by the decoder, making the training computational time prohibitive (for $D = 256$, choosing $K \geq 2$ implies a Fisher vector's dimension greater than a thousand).

**VLAD encoding** In this case, the context vectors are represented as Vector of Locally Aggregated Descriptors (VLAD, Jégou et al. (2010); Arandjelovic & Zisserman (2013)). VLAD is a feature encoding and pooling method, similar to Fisher Vectors. It encodes a set of local feature descriptors extracted from a sequence and denoted as $W = (w_1, \ldots, w_n : w \in \mathbb{R}^D)$, using a clustering method such as K-means. Let $q_{ik}$ be the hard assignment of data vectors $w_i$ to the $k^{th}$ cluster, such that $q_{ik} \geq 0$ and $\sum_{k=1}^{K} q_{ik} = 1$. Furthermore, the assignments of features to dictionary elements must be pre-computed, for example by using KD-trees (Beis & Lowe, 1997; Silpa-Anan & Hartley, 2008). VLAD encodes features $W$ by considering the residuals

$$
v_k = \sum_{i=1}^{N} q_{ik}(w_i - \mu_k),
\tag{8}
$$

where $\mu_k \in \mathbb{R}^N$ is the $k^{th}$ cluster centre (or mean). Then, these residuals are stacked together to obtain the final context vector.

**GeoSeq2Seq** Our novel contribution is to couple a Seq2Seq model with information geometric representation of the context vector. To do so, we start with a general Seq2Seq trained model. For each source-destination tuple, we use this encoder to compute the context vectors, which have identical same size (i.e., either 256 or 512 as specified in the Results section). Then, we train the GMM on the context vectors obtained from the training sequences and finally use the means and variances to construct the Fisher Vectors. Similarly, for the VLAD-based approach, we train the centres and assignments from K-means and KD-trees on the context vectors obtained from the training sequences. We then use them to generate the VLAD encoding. Subsequently, we use the new context vectors encoded with FV/VLAD as an initial hidden state to train the decoder, finally providing us with the shortest path as output.

As the work of Sydorov et al. (2014) suggests it is possible to learn the Fisher kernel parameters in an end-to-end manner for a convolutional neural network. A future development for this current work will be to inquire whether end-to-end training of Fisher encoding for Seq2Seq models can be attained.

## 4 RESULTS

### 4.1 SHORTEST PATH PROBLEM

For the graph of Minnesota with $n = 376$ nodes and 455 edges, we generated 3,000 shortest routes between two nodes picking them uniformly at random) and using the $A^*$ algorithm with a heuristic based on the Euclidean distance. We used these routes as the training set for the Seq2Seq algorithms using a 67-33% training-test splits; we train the network for 400 epochs, updating the parameters with an Adam optimisation scheme (Kingma & Ba, 2014), with parameters $\beta_1 = 0.9$ and $\beta_2 = 0.999$, starting from a learning rate equal to $10^{-3}$. Since Fisher encoding requires a double length for the context vector (it considers means and covariances, see Eqn. (5)), we compared it to a basic Seq2Seq with 256 and 512 hidden units. On the other hand, VLAD encodes only the residual (see Eqn. (8)), therefore we instantiate it with 256 and 512 hidden units.

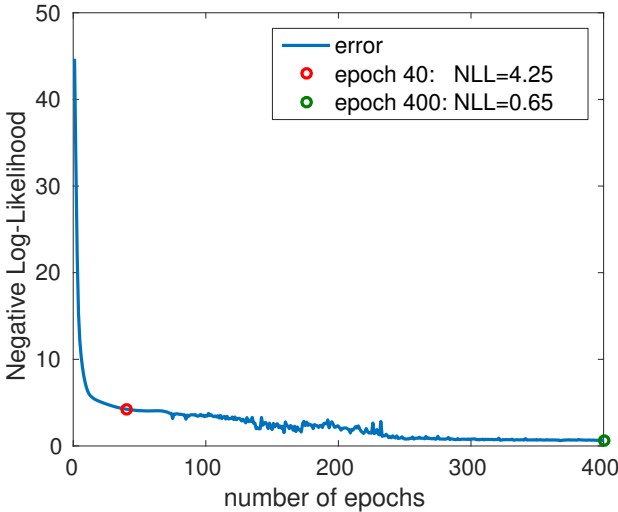

Figure 2: **Training error.** We illustrate the negative log-likelihood loss function during the training phase. The adjacency matrix is iteratively learnt during such a training phase. We highlight the situation after 40 epochs (red dot) and at the end of the training (after 400 epochs, green dot) when the back-propagation through time algorithm has converged. The paths are shown in Figure 3.

Moreover, in the decoder, the following node in the predicted sequence is computed choosing among all the other $n - 1$ nodes from a softmax distribution. If we stop the training early, for example, after only 40 epochs (see Figure 2), the network cannot reproduce possible paths (see Figure 3 on the left-hand side). On the other hand, if we train the network for more epochs, the training converges after 400 epochs (green dot in Figure 2). Therefore, as shown in Figure 3 (right), if we compute two paths, one starting from the source (magenta dash-dotted line) and the other from the destination (cyan), intersecting them allows us to predict the shortest path (red). This is equal to the one generated by $A^*$ (green). This means that the network is capable of learning the adjacency matrix of the graph.

Then, comparing different approaches, the prediction accuracy on the test data-set is reported in Table 1. As we can see, for what concerns the RNN decoder, doubling the hidden state dimension marginally increases the percentage of shortest paths (1%) and the successful paths, that are not necessarily the shortest (0.2% and 1.6% for GRU and LSTM encoders, respectively). Our proposed information geometry based Fisher encoding achieves an increased accuracy in finding the shortest paths (56% and 60% for LSTM and GRU, respectively). Furthermore, if a VLAD encoding is employed, GRU networks have a higher approximation capability with more than 65% of accuracy on the shortest paths and 83% on the successful cases. Similarly, if the decoder is an LSTM, the accuracies on the shortest paths of simple Seq2Seq models are around 50%, while our *GeoSeq2Seq* models with Fisher encoding we get close to 60% and even above with the VLAD encoding, achieving 65% (and 82% of successful paths) in the case of the LSTM encoder. This means that our probabilistic representation of the latent embedding supersedes the non-probabilistic one by 10-15%.

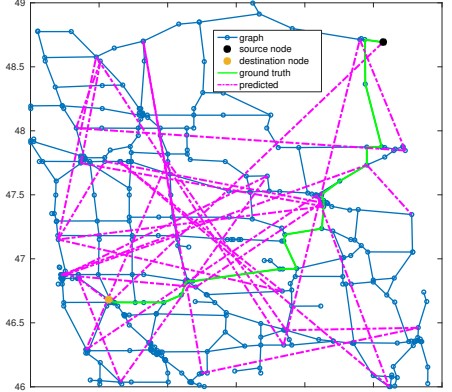 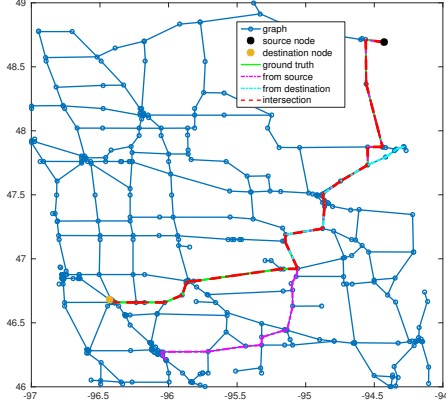

Figure 3: **Predicted path.** Here, we show an example of a prediction of shortest path on the Minnesota dataset (blue graph). The $A^*$ shortest path between the source (black dot) and destination (yellow dot) nodes is represented in green. On the left-hand side, we show the prediction (magenta) after only 40 epochs of training (Figure 2). On the right-hand side, instead, we show the prediction at the end of the training: we compute two paths, one starting from the source (magenta) and the other from the destination (cyan), and finally, we intersect them to compute the predicted shortest path (red). The 'flying-object syndrome' only occurs during the earlier phase of training.

| Method | Shortest | Successful |
|---|---|---|
| LSTM2RNN (256) | 47% | 69.5% |
| LSTM2RNN (512) | 48% | 71.1% |
| GRU2RNN (256) | 48.3% | 73.1% |
| GRU2RNN (512) | 49% | 73.3% |
| | | |
| LSTM2RNN with FV | 56% | 76.4% |
| GRU2RNN with FV | 60.1% | 79% |
| | | |
| LSTM2RNN with VLAD (256) | 59.3% | 76.8% |
| LSTM2RNN with VLAD (512) | 60.1% | 76.2% |
| GRU2RNN with VLAD (256) | 64% | 79.9% |
| GRU2RNN with VLAD (512) | **65.7%** | **83%** |
| | | |
| LSTM2LSTM (256) | 50.3% | 72.7% |
| LSTM2LSTM (512) | 54% | 73.3% |
| GRU2LSTM (256) | 48% | 69.5% |
| GRU2LSTM (512) | 48.2% | 73.5% |
| | | |
| LSTM2LSTM with FV | 57.9% | 78.8% |
| GRU2LSTM with FV | 59% | 79.3% |
| | | |
| LSTM2LSTM with VLAD (256) | 62.4% | 79.8% |
| LSTM2LSTM with VLAD (512) | **65.2%** | **82.1%** |
| GRU2LSTM with VLAD (256) | 63% | 80.6% |
| GRU2LSTM with VLAD (512) | 63.7% | 81.6% |

Table 1: **Results on the Minnesota graph.** Percentage of shortest path and successful paths (that are not necessarily shortest) are shown for a wide-variety of Seq2Seq models, with context vector dimension equal to either 256 or 512. All scores are relative to an $A^*$ algorithm, that achieves a shortest path score of 100%.

## 4.2 NEURAL TURING MACHINES TASKS

In order to provide more generality to our method, in this section, we apply the GeoSeq2Seq model to solve algorithmic tasks similar to those on which Neural Turing Machines (NTM) were evaluated Graves et al. (2014), albeit without a need for an external memory module. Specifically, we present results for a simple algorithmic task such as copying, and a more complex semantic task such as associative recall. For these tasks, we used an LSTM2LSTM with VLAD encoding and followed a configuration of hyper-parameters similar to the one in Graves et al. (2014) i.e. we used RMSProp algorithm with momentum set to 0.9, a learning rate equal to 1e-4 and clipped the gradient during the backpropagation to the range (-10,10).

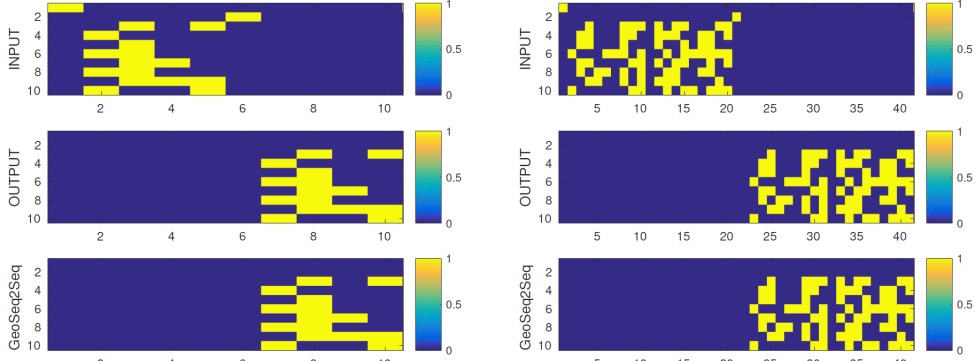

Figure 4: **Copying task.** Two results for the copying task. On the left-hand side (a), we considered a binary sequence of length 4 bounded by the start flag and the stop one. The sequence is correctly repeated as output by our GeoSeq2Seq method. On the right-hand side (b), instead, we show a longer sequence (i.e. of length 19), which is correctly copied too.

**Copying task** The copying task consists in remembering an eight bit random binary input sequence between a start flag and a stop flag and copying it in the output sequence after the stop flag (see Figure 4). As we can see, our method can copy both shorter (Figure 4a) and longer sequences (Figure 4b). We run 1e5 experiments as test sequences, with random lengths between 2 and 20. Just like NTMs all of them were reproduced at the output.

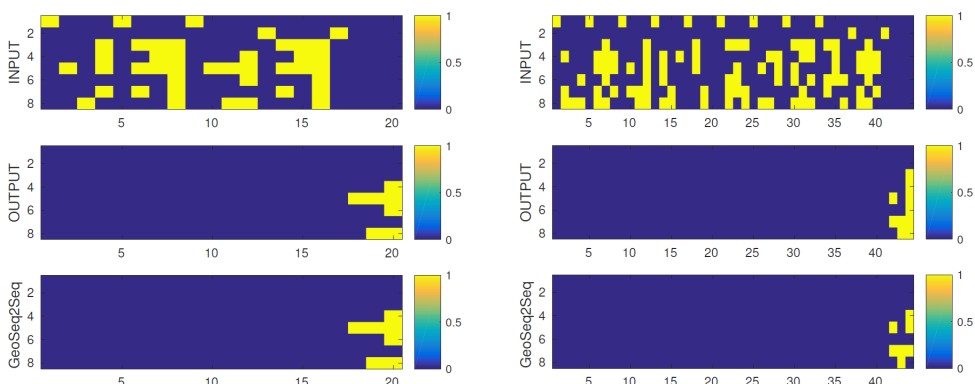

Figure 5: **Associative recall task.** Two results for the associative recall task. On the left-hand side (a), we considered a sequence of 3 items, with the query of the second one. The third item is correctly repeated as output by our GeoSeq2Seq method. On the right-hand side (b), instead, we show a longer sequence (i.e. 9 items) with the query of the second one. The third item is repeated almost correctly: the first two columns are correct, while the last one misses some components.

**Associative recall task** The associative recall task consists in remembering an input sequence of item bounded on the left and right by a delimiter flag and repeating the item following the one

between the two query flags in the output sequence (see Figure 5). We define an item as a sequence of three random binary columns. For 1e5 test experiments with a random length of the sequence between 2 and 12 items, we achieve an accuracy equal to 13.39%. This apparently low accuracy is because, as we can see, our GeoSeq2Seq can reproduce the correct next item (Figure 5a) for shorter sequences, while it may fail in few components when the sequence becomes longer (Figure 5b), which is considered an error, as well.

## 5 DISCUSSION

This paper proposes the *GeoSeq2Seq*, an information geometric Seq2Seq architecture that utilises an information geometric embedding of the context vector. The RNN is tasked with learning the adjacency matrix of the graph such that at each decision step the search space becomes $n - 1$, where $n$ is the total number of nodes in the graph. Unlike algorithms like *q-routing* (Boyan & Littman, 1994) that constrains the search space to contain only the connected neighbours, our instantiation of the Seq2Seq operates under a larger search space. Indeed, the accuracy of our algorithm is expected to increase where we use neighbourhood information from the adjacency matrix. In summary, such a recurrent network shows increased fidelity for approximating the shortest route produced by a meta-heuristic algorithm.

Apart from encoding, context vector stacking using dual encoders have been proven to be beneficial (Bay & Sengupta, 2018). Utilising homotopy based continuation has been a different line of work where the emphasis lies in smoothing the loss function of the recurrent network by convolving it with a Gaussian kernel (Bay & Sengupta, 2017). All of these strategies have shown to improve the temporal memory of the recurrent network. This line of work is distinct from architecture engineering, that has placed emphasis on constructing exquisite memory mechanism for the recurrent networks. For example, Neural Turing Machines (Graves et al., 2014) are augmented recurrent networks that have a (differentiable) external memory that can be selectively read or written to, enabling the network to store the latent structure of the sequence. Attention networks, on the other hand, enable the recurrent networks to attend to snippets of their inputs (*cf.* in conversational modelling Vinyals & Le (2015)). Similarly, it is also possible to embed the nodes of the graph as preprocessing step using methods such as – (a) DeepWalk (Perozzi et al., 2014), that uses local information obtained from truncated random walks to encode social relations by treating walks as the equivalent of sentences, (b) LINE (Tang et al., 2015), which optimises an objective function preserving both local and global structures, and (c) node2vec (Grover & Leskovec, 2016), that learns continuous feature representations for nodes in networks. Nevertheless, for the long-term dependencies that shortest paths in large graphs inherit, these methods are small steps towards alleviating the central problem of controlling spectral radius in recurrent neural networks.

The 4-5% gain in accuracy for VLAD in contrast to Fisher encoding is a bit surprising. As a matter of fact, Fisher encoding should be more precise since it takes covariances into account. We notice that the covariances matrix (instantiated as a diagonal matrix due to computational constraint) has a high condition number. The condition number $\kappa$ of a matrix $M$ is defined as $\kappa(M) = \|M\|\|M^{-1}\|$, or equivalently it is the ratio of the largest singular value of that matrix to its smallest singular value (Belsley et al., 2005). Practically, it measures how much small variation in the input argument can propagate to the output value. In particular, we obtain a condition number equal to 2.69e3 and 1.70e3 for the LSTM and GRU encoders, respectively. We believe it is for this reason that the GRU encoder leads to a better accuracy than the LSTM encoder, as shown in Table 1.

The use of Riemann geometry to encode context (feature) vector has a long history in computer vision (Srivastava & Turaga, 2015), our work demonstrates yet another way to embed the curved geometry of the context vector for decoding. The Riemannian metric for a recurrent network can be evaluated in two ways – one where we describe the probability distribution over the entire sequence and another where we describe a conditional distribution at time $i$ conditioned on time $i - 1$. We anticipate that the latter is more suited to a dynamic scenario (where the structure of the graph may be slowly changing) while the former is more suitable for static graphs. Analytically, averaging over time and assuming ergodicity, both metric should be fairly close to one another, nonetheless, it is only with further experiments we can demonstrate the value of one over the other.

In this paper, we have constrained the use of the Riemannian metric on the encoding end, one can follow a similar treatise for the decoder. Together with architecture engineering (Graves et al.,

2014), natural gradient-based optimization (Pascanu & Bengio, 2013), homotopy continuation (Bay & Sengupta, 2017) and ensembling of recurrent encoders (Bay & Sengupta, 2018), we posit that understanding the information geometry of recurrent networks can take us a step closer to finessing the temporal footprint of a sequence learning network.

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
