# OpenReview forum: "GeoSeq2Seq: Information Geometric Sequence-to-Sequence Networks"
_ICLR.cc/2018/Conference — Invite to Workshop Track_

### Official Review · AnonReviewer2 · 2017-11-25
**Missing references and insufficient experiments**

**Rating:** 4
**Confidence:** 4

**Review:**

The paper proposes a method for augmenting sequence-to-sequence (seq2seq) methods with Fisher vector encodings, allowing the decoder to better model the geometric structure of the embedding space. Experiments are performed on a shortest-route problem, where augmenting standard seq2seq architectures with Fisher vectors improves performance.

Pros:
- Combining deep learning with methods from information geometry is an interesting direction for research
- Method is a generic drop-in replacement for improving any seq2seq architecture
- Experimental results show modest performance improvements over vanilla seq2seq methods

Cons:
- Missing references for prior work combining information geometry and deep learning
- Insufficient explanation of the method
- Only experimental results are a nonstandard route-finding task
- Missing references and baselines for prior work on deep learning on graphs

The general research direction of combining deep learning with methods from information geometry is an exciting and fertile area for interesting work. Unfortunately this paper fails to cite or discuss much recent work in this area; for example natural gradient methods in deep learning have recently been explored in [1, 2, 3]; more closely related to the topic of this paper, [4] and [5] have combined Fisher vector encodings and deep networks for image classification tasks. Although these prior methods do not consider the use of recurrent networks, the authors should discuss how their method compares to the approaches of [4] and [5].

The method is not described in sufficient detail. How exactly is the Fisher encoding combined with the recurrent neural network? In particular, how is GMM fitting interleaved with learning the RNN? Do you backpropagate through the GMM fitting procedure in order to jointly learn the RNN parameters and the GMM for computing Fisher encodings? Or is GMM fitting an offline step done once, after which the RNN decoder is learned on top of the Fisher encodings? The paper should clarify along these points. As a side note, it also feels a little disingenuous to describe the method in terms of GMMs, but to perform all experiments with K=1 mixture components; in this setting the GMM degrades to a simple Gaussian distribution.

The proposed method could in theory be used as a drop-in replacement for seq2seq on any task. Given its generality, I am surprised at the nonstandard choice of route-finding in a graph of Minnesota roads as the only task on which the method is tested; as a minimum the method should have been tested on more than one graph.

More generally, I would have liked to see the method evaluated on multiple tasks, and on more well-established seq2seq tasks so that the method could be more easily compared with previously published work. Strong results on machine translation would be particularly convincing; the authors might also consider algorithmic tasks such as copying, repeat copying, sorting, etc. similar to those on which Neural Turing Machines were evaluated.

I am not sure that seq2seq is the best approach for the route-finding task. In particular, since the input is encoded as a [source, destination] tuple it has a fixed length; this means that you could use a feedforward rather than recurrent encoder.

The paper also fails to cite or discuss recent work involving deep learning on graphs. For example Pointer Networks [6] use a seq2seq model with attention to solve convex hull, Delaunnay Triangulation, and traveling salesman problems; however Pointer Networks assume that the entire graph is provided as input to the model, while in this paper the network learns to specialize to a single graph. In that case, the authors might consider embedding the nodes of the graph using methods such as DeepWalk [7], LINE [8], or node2vec [9] as a preprocessing step rather than learning these embeddings from scratch.

From Table 1, seq2seq + VLAD significantly outperforms seq2seq + FV. Given these results, are there any reasons why one should use seq2seq + FV instead of seq2seq + VLAD?

Overall I think that this paper has some interesting ideas. However, due to a number of missing references, unclear description of the method, and limited experimental results I feel that the paper is not ready for publication in its current form.


References

[1] Grosse and Salakhutdinov, “Scaling Up Natural Gradient by Sparsely Factorizing the Inverse Fisher Matrix”, ICML 2015

[2] Grosse and Martens, “A Kronecker-factored approximate Fisher matrix for convolution layers”, ICML 2016

[3] Desjardins et al, “Natural Neural Networks”, NIPS 2015

[4] Simonyan et al, “Deep Fisher Networks for Large-Scale Image Classification”, NIPS 2013

[5] Sydorov et al, “Deep Fisher Kernels - End to End Learning of the Fisher Kernel GMM Parameters”, CVPR 2014

[6] Vinyals et al, “Pointer Networks”, NIPS 2015

[7] Perozzi et al, “DeepWalk: Online Learning of Social Representations”, KDD 2014

[8] Tang et al, “LINE: Large-scale Information Network Embedding”, WWW 2015

[9] Grover and Leskovec, “node2vec: Scalable Feature Learning for Networks”, KDD 2016

---

> ### Author Response · Authors · 2018-01-05
> **Response to reviewer 3**
>
> We thank the Reviewer for his/her comments and for providing useful feedback. Since, we were constrained on time and resources (to train on a machine translation task), in our updated version, we have now introduced two benchmark problems such as copying and recalling sequences. On both problems, the GeoSeq2Seq network has been as successful as the Neural Turing Machine, albeit without a need for an external memory module.
>
> We have also added a related work section as well as additional information for model construction. In our related work section, we have discussed all of the missing prior work suggested by the Reviewer including, natural neural networks, Fisher vectors for image classification, pointer networks, etc.
>
> As the work of Sydorov et al. (2014) suggests it is indeed possible to learn the Fisher kernel parameters in an end-to-end manner. In our current work, we have first used the latent vectors learnt using a vanilla Seq2Seq training process to initiate a GMM, and thereof the Fisher kernel, subsequently a decoder is trained on the Fisher kernel to generate a prediction.
>
> Indeed, we agree with the reviewer that a Seq2Seq network may not be the best approach for route finding task; our motivation to use a route finding problem is to enable us to control task difficulty, and not for replacing algorithms such as meta-heuristics, integer programming, etc.
>
> The idea to embed the nodes of the graph using methods like node2vec, DeepWalk and LINE are very useful and we anticipate them to finesse the accuracy of the GeoSeq2Seq. We would undoubtedly explore this avenue for our future work.
>
> We have now explained why seq2seq + VLAD significantly outperforms seq2seq + FV and therefore it is preferable to use it instead of seq2seq+FV. We anticipate the performance being directly related to the condition number (the ratio of the largest to smallest singular value in the singular value decomposition of a matrix) of the Fisher Information Matrix.

---

### Official Review · AnonReviewer1 · 2017-11-27
**Missing some connections**

**Rating:** 5
**Confidence:** 2

**Review:**

==== UPDATE AFTER REVIEWER RESPONSE

I apologize to the authors for my late response.

I appreciate the reviewer responses, and they are helpful on a number of
fronts. Still, there are several problematic points.

First, as the authors anticipated, I question whether the geometric encoding
operations can be included in an end-to-end learning setting. I can imagine
several arguments why an end-to-end algorithm may not be preferred, but the
authors do not offer any such arguments.

Second, I am still interested in more discussion of the empirical investigation
into the behavior of the algorithm. For example, "Shortest" and "Successful"
in Table 1 still do not really capture how close "successful but not shortest"
paths are to optimal.

The authors have addressed a number of my concerns, but there
are a few outstanding concerns. Also, other reviewers are much more familiar
with the work than myself. I defer to their judgement after the updates.

==== Original review

In this work, the authors propose an approach to adapt latent representations to account for local geometry in the embedding space. They show modest improvement compared to reasonable baselines.

While I find the idea of incorporating information geometry into embeddings very promising, the current work omits a number of key details that would allow the reader to draw deeper connections between the two (specific comments below). Additionally, the experiments are not particularly insightful.

I believe a substantially revised version of the paper could address most of my concerns; still, I find the current version too preliminary for publication.

=== Major comments / questions

The transformation from context vectors into Fisher vectors is not clear. Presumably, shortest paths in the training data have different lengths, and thus produce different numbers of context vectors. Does the GMM treat all of these independently (regardless of sample)? or is a separate GMM somehow trained for each training sequence? The same question applies to the VLAD-based approach.

In a related vein, it is not clear to what extent this method depends on the sequential nature of the considered networks. In particular, could a similar approach be applied to latent space embeddings from non-sequential models?

It is not clear if the geometric encoding operations are differentiable, or more generally, the entire training algorithm is not clear.

The choice to limit the road network graph feels quite arbitrary. Why was this done?

Deep models are known to be sensitive to the choice of hyperparameters. How were these chosen? was a validation set used in addtion to the training and testing sets?

The target for training is very unclear. Throughout Sections 1 and 2, the aim of the paper appears to be to learn shortest paths; however, Section 3 states that the “network is capable of learning the adjacency matrix”, and the caption for Figure 2 suggests that “[t]he adjacency matrix is iteratively learnt (sic)....” However, calculating such training error for back-propagation/optimization would seem to rely on *already knowing* the adjacency matrix.

The performed experiments are minimal and offer very little insight into what is learned. For example, does the model predict “short” shortest paths better than longer ones? what do the “valid but not optimal” paths look like? are they close to optimal? what do the invalid paths look like? does it seem to learn parts of the road network better than others? sparse parts of the network? dense parts?

=== Minor comments / questions

The term “context vector” is not explicitly defined or described. Based on the second paragraph in the “Fisher encoding” section, I assume these are the latent states for each element in the shortest path sequences.

Is the graph directed? weighted? by Euclidean distance? (Roads are not necessarily straight, so the Euclidean distance from intersection to intersection may not accurately reflect the distance in some cases.)

Are the nodes sampled uniformly at random for creating the training data?

Is the choice to use a diagonal covariance matrix (as opposed to some more flexible one) a computational choice? or does the theory justify this choice?

Roughly, what are the computational resources required for training?

The discussion should explain “condition number” in more detail.

Do the “more precise” results for the Fisher encoding somehow rely on an infinite mixture? or, how much does using only a single component in the GMM affect the results?

It is not clear what “features” and “dictionary elements” are in the context of VLAD.

What value of k was used for K-means clustering for VLAD?

It is not possible to assess the statistical significance of the presented experimental results. More datasets (or different parts of the road network) or cross-validation should be used to provide an indication of the variance of each method.

=== Typos, etc.

The paper includes a number of runon sentences and other small grammatical mistakes. I have included some below.

The first paragraph in Section 2.2 in particular needs to be edited.

The references are inconsistently and improperly (e.g., “Turing” should be capitalized) formatted.

It seems like that $q_{ik} \in \{0,1\}$ for the hard assignments in clustering.

---

> ### Author Response · Authors · 2018-01-05
> **Response to reviewer 2**
>
> We thank the Reviewer for his/her comments and for providing useful feedback. We have now included further information for constructing the Fisher Vectors, along with the relevant references in computer vision, where it has been routinely utilised.
>
> Shortest paths in the training data do have different lengths, but we build one context vector for each source-destination tuple and all context vectors have the same fixed length (we choose either 256 or 512). We then train the GMM on the context vectors obtained from the training sequences and finally use the means and variances to build the Fisher vectors. Similarly, for the VLAD-based approach, we train the centers and assignments from K-means and KD-trees on the context vectors obtained from the training sequences and use them to generate the VLAD encoding.
>
> The formulation of the Fisher information metric, in its current form, can be directly applied to latent space embeddings from non-sequential models. However, as mentioned in the discussion: The Riemannian metric for a recurrent network can be evaluated in two ways -- one where we describe the probability distribution over the entire sequence and another where we describe a conditional distribution at time i conditioned on time i-1. We anticipate that the latter is more suited to a dynamic scenario (where the structure of the graph may be slowly changing) while the former is more suitable for static graphs. Analytically, averaging over time and assuming ergodicity, both metric should be fairly close to one another, nonetheless, it is only with further experiments we can demonstrate the value of one over the other.
>
> It is a very good question whether the geometric encoding operations are differentiable. We anticipate this goes on to enquire if end-to-end training of Fisher encoding for Seq2Seq models can be attained. Infact Sydorov et al. (2014) have shown just this for convolutional neural networks.
>
> The selection of the road network graph was not arbitrary. In fact,  our motivation for using a route finding problem is to enable us to control task difficulty, and not for replacing algorithms such as meta-heuristics, integer programming, etc. In our updated version, we have added two other algorithmic tasks – copying and associative recall of sequences.
>
> In the interest of time and resources, we have not utilised hyper-parameter optimization for this paper. In future, with more compute resources, we anticipate utilizing Bayesian optimization to obtain hyperparameters during the validation phase. Also, we were constrained by time and resources (to train on a machine translation task), in our updated version, we have now introduced two benchmark problems such as copying and recalling sequences. On both problems, the GeoSeq2Seq network has been as successful as the Neural Turing Machine, albeit without a need for an external memory module. Again due to limited computation resources, we had to limit our experiments to a diagonal covariance matrix and K=1 as the number of components of the Gaussian Mixture Model. There is no reason why a full covariance matrix or increases mixture components cannot be used.
>
> Of course, finding shortest routes consisting of few nodes is easier. Since the training set is built by sampling source and destination nodes uniformly at random, we can have routes with many nodes and our algorithm can reproduce the shortest path correctly (see for example Figure 3b). It is also possible to find a route between source and destination nodes that is not the shortest (i.e. the sum of the distances between the nodes in the path is greater than the ground truth one). We reported also the accuracy in this case, because in some application it may be enough to reach the destination even if the path is not the shortest. “Invalid” paths are routes that diverge in wrong directions and do not reach the destination.
>
> The considered graph is undirected and weighted by Euclidean distance. Roads may not be straight but they can be approximated by straight parts from intersection (node) to intersection (node).
>
> We have now included a formal definition of the condition number in the main text.
>
> Sydorov et al, “Deep Fisher Kernels - End to End Learning of the Fisher Kernel GMM Parameters”, CVPR 2014

---

### Official Review · AnonReviewer3 · 2017-12-03
**Incremental contribution and detail missing.**

**Rating:** 5
**Confidence:** 4

**Review:**

In this paper, the authors propose to integrate the Fisher information metric with the  Seq2Seq network, which abridges the gap between deep recurrent neural networks and information geometry. By considering of the information geometry of the latent embedding, the authors propose to encode the RNN feature as a Fisher kernel of a parametric Gaussian Mixture Model, which demonstrate an experimental improvements compared with the non-probabilistic embedding.

The idea is interesting. However, the technical contribution is rather incremental. The authors seem to integrate some well-explored techniques, with little consideration of the specific challenges. Moreover, the experimental section is rather insufficient. The results on road network graph is not a strong support for the Seq2Seq model application.

---

> ### Author Response · Authors · 2018-01-05
> **Response to reviewer 1**
>
> We thank the Reviewer for his/her comments. The specific challenge that this paper sets to achieve is to come up with a methodology to increase the temporal memory of a recurrent neural network. In order to achieve this, we use utilise the 2nd order geometry of the latent embeddings, instead of invoking an external memory unit, as in the Neural Turing Machine. A road network graph gives us a straightforward way to control the length of temporal information required to be stored by the neural network. An additional benefit of using the road network graph is unlike other benchmark toy problems, a solution to the shortest route problem on graphs not only had an illustrious history but is also a real-life scenario. In our updated version, we have now introduced two benchmark problems such as copying and recalling sequences. We have also added a related work section as well as additional information for model construction.

---

### Decision · Program_Chairs · 2018-01-29
**ICLR 2018 Conference Acceptance Decision**

**Decision:**

Invite to Workshop Track

**Comment:**

The reviewers found the paper meaningful but noted that they were not convinced by the experiments as they stand and the presentation was dense for them.